# MIRA: Multi-view Information Retrieval with Adaptive Routing for Test-time Long-video Comprehension

**Zecheng Hao**[†]                                              *haozecheng@pku.edu.cn*
*School of Computer Science, Peking University*
*State Key Laboratory for Multimedia Information Processing, Peking University*

**Wenxuan Ma**[†]                                              *wenxuanbit@gmail.com*
*Beijing Academy of Artificial Intelligence (BAAI)*

**Yufeng Cui**                                              *yfcui@baai.ac.cn*
*Beijing Academy of Artificial Intelligence (BAAI)*

**Shuang Li**                                              *shuangliai@buaa.edu.cn*
*School of Artificial Intelligence, Beihang University*

**Xinlong Wang**                                              *xinlong.wang96@gmail.com*
*Beijing Academy of Artificial Intelligence (BAAI)*

**Tiejun Huang**                                              *tjhuang@pku.edu.cn*
*School of Computer Science, Peking University*
*State Key Laboratory for Multimedia Information Processing, Peking University*
*Beijing Academy of Artificial Intelligence (BAAI)*

**Reviewed on OpenReview:** *https://openreview.net/forum?id=LZb2kzO8tu*

## Abstract

Foundational Multi-modal Large Language Models (MLLMs) have achieved rapid progress in handling complex tasks across diverse modalities. However, they still struggle to deliver satisfactory performance on Long-video Comprehension (LVC) tasks involving thousands of frames. Existing optimization strategies can be broadly categorized into LVC-specific fine-tuning, built-in token compression and training-free keyframe extraction, with the latter being most suitable for flexible deployment across various MLLMs. Unfortunately, current training-free approaches predominantly focus on query-frame relevance retrieval, overlooking other levels of visual information and the inherent heterogeneity of LVC tasks. In this work, we propose the **M**ulti-view **I**nformation **R**etrieval with **A**daptive Routing (**MIRA**) framework, which evaluates video frames using distinct metrics for relevance and causality, combines these scores to select a balanced pool of keyframes, and employs an adaptive feedback loop to tailor the retrieval process to different user queries, enabling more precise and sample-grained video comprehension. Extensive experiments demonstrate the advanced performance of our scheme across multiple challenging LVC benchmarks. For instance, integrating **MIRA** with Qwen-2.5-VL yields performance gains of 3.5% to 13.1% on LVB, VideoMME and MLVU.

## 1 Introduction

Long-video Comprehension (LVC), as one of the most challenging multi-modal reasoning tasks, has emerged as a critical benchmark for evaluating the advanced capabilities of Multi-modal Large Language Model

---

[†]Equal contributions for this work.

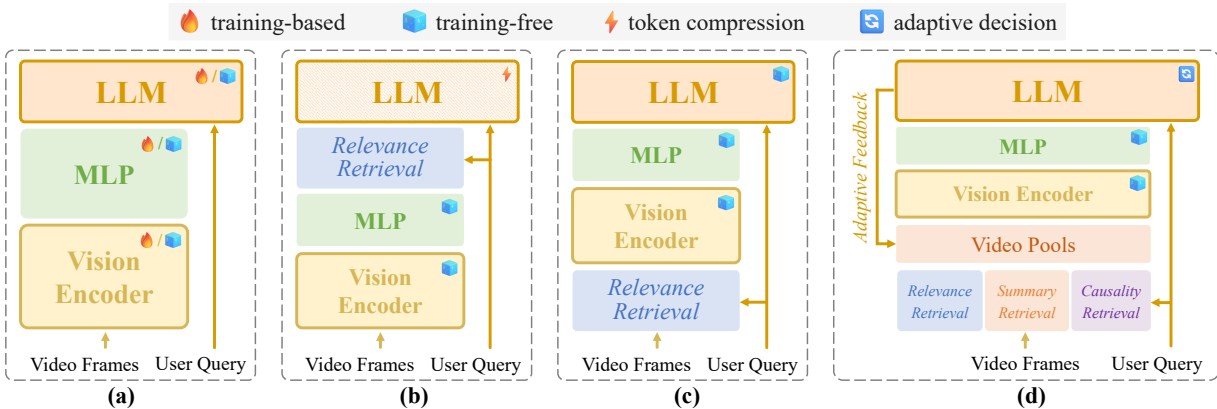

Figure 1: Comparison of different optimization schemes for LVC. (a): feeding uniformly-sampled video frames into (fine-tuned) MLLMs; (b): built-in visual token compression for MLLMs; (c): performing relevance retrieval for the video and feeding the obtained keyframe set into MLLMs; (d): **MIRA** framework mixes keyframe sets obtained from multiple perspectives and feeds them into MLLMs, with the specific blending ratio (video pool) regulated by the model's adaptive feedback.

(MLLM). While current foundational MLLMs (Zhang et al., 2024b; Bai et al., 2025; Chen et al., 2024b) demonstrate strong generalization across diverse multi-modal tasks, they still struggle when directly applied to complex or fine-grained reasoning tasks grounded in long-form videos. To address this limitation, as illustrated in Fig.1(a)-(c), prior works have explored three primary routes: first, researchers have attempted to train specialized MLLMs or fine-tune components of existing models specifically for long-video processing (Chen et al., 2024a; Shen et al., 2024; Zohar et al., 2025; Islam et al., 2025). However, due to the rich content diversity and enormous visual token budget inherent in long videos, such approaches demand prohibitively high training overhead. Moreover, the scarcity of supervised fine-tuning data often fails to adequately cover the full spectrum of video semantics, limiting the generalizability of these improved models.

Furthermore, some efforts have also focused on integrating token compression techniques within MLLMs to expand their effective context capacity (Cheng et al., 2025; Wang et al., 2025a; Gao et al., 2025), thereby enabling the model to ingest more visual frames. Yet, since different MLLMs employ heterogeneous architectures, these compression methods may lack cross-model compatibility, constraining their broad applicability. In addition, and most recently, a promising line of research has proposed externalizing keyframe extraction as a training-free pre-processing pipeline by decoupling it from the MLLM's core architecture (Tang et al., 2025b; Ye et al., 2025). This approach leverages the strengths of pre-trained models while remaining architecture-agnostic (Wang et al., 2025b; Xu et al., 2025), thereby gaining increasing attention in the community.

However, current keyframe extraction methods predominantly use CLIP-like models (Radford et al., 2021) for relevance scoring, which tend to concentrate high scores on narrow temporal segments. Consequently, the resulting keyframes often fail to provide sufficient coverage of the event's full scope and context. More importantly, we argue that true understanding requires more than just relevance; it requires capturing the causal context – the frames that explain why a moment is significant. These crucial pieces of evidence are largely ignored by current methods. Furthermore, this one-size-fits-all approach fails to adapt to the user's query. A query about a fine-grained detail needs a different set of keyframes than one about a complex event, a distinction overlooked in prior works that should dictate whether the extraction is localized or distributed.

In this work, we attempt to address the above limitations through the following steps, as shown in Fig.1(d): (i) beyond direct relevance-based retrieval, we further construct distinct keyframe sets from the perspectives of scene relevance and causality; (ii) these keyframe sets, derived from heterogeneous sources, are then blended at multiple ratios to generate diverse video sampling pools in batch; (iii) leveraging the adaptive feedback capability of MLLMs, we enable fine-grained exploration over these video pools. Our contributions are summarized as follows:

- We introduce a novel keyframe selection framework that, unlike prior works centered on a singular scoring metric, uniquely integrates three complementary scoring dimensions: global-regional relevance, hierarchical scene context, and causal relationships. This enables our model to capture a holistic understanding of video content—recognizing not only what is relevant, but also its broader context and causal significance.

- We establish the novel insight that by blending our three scoring metrics in varying ratios, we can construct distinct keyframe "pool" that are inherently specialized for different query intents (e.g., descriptive vs. causal questions). This reframes the challenge from generic frame selection to targeted, query-aware sampling.

- We design a novel, closed-loop adaptive routing strategy that leverages a model feedback loop. Instead of using a static, one-size-fits-all approach, this mechanism dynamically analyzes a user's query and selects the optimal keyframe pool on-the-fly, intelligently adapting its retrieval focus between relevance, context, and causality to best answer the specific question.

- Our MIRA framework significantly outperforms a wide range of leading proprietary and open-source models, achieving state-of-the-art results on challenging video understanding benchmarks.

## 2 Related Works

**Mainstream Foundational MLLMs.** Recent advances in foundational MLLMs, including model families such as GPT-4o (Hurst et al., 2024), LLaVA (Li et al., 2024; Zhang et al., 2024b), Qwen-VL (Bai et al., 2025), InternVL (Chen et al., 2024b) and NVILA (Liu et al., 2025b), have demonstrated remarkable capabilities across diverse multi-modal tasks. These models are typically trained through multi-stage Pre-training and Instruction Fine-tuning on heterogeneous data modalities, including image, text, video, chart, document, mathematical expressions and Graphical User Interfaces (GUIs), enabling them to perform complex operations such as visual question answering, cross-modal grounding, temporal localization and structured reasoning. Despite their broad generalization power, when applied to LVC, these models still exhibit a substantial performance gap relative to human-level evaluation when simply applying uniform frame sampling and directly feeding the resulting frame sequence.

**Training-based Models for LVC.** Supervised Fine-tuning (SFT) of MLLMs remains a dominant strategy for enhancing the reasoning capability and performance on LVC tasks (Chen et al., 2024a; Shen et al., 2024). Frame-Voyager (Yu et al., 2024) leverages the prediction loss of a pre-trained MLLM to collect high-quality training data from diverse frame combinations, enabling the learning of an automated scoring component for keyframe selection. Hu et al. (2025b) annotate video samples with dual pseudo labels (spatial and temporal) to train a lightweight frame selector tailored for LVC. Zohar et al. (2025) systematically distill empirical guidelines for model training and inference in the domain of LVC, based on which they propose the family of Apollo models that can effectively address long-range temporal understanding. In addition, BIMBA (Islam et al., 2025) employs a state-space model with selective scan mechanisms to dynamically transmit only those most informative tokens to the language decoder. Similarly, ViLaMP (Cheng et al., 2025) integrates differential keyframe selection with weighted feature fusion, significantly suppressing temporal redundancy while retaining critical visual semantics. It is worth noting that recent works have further explored the integration of Reinforcement Learning (RL) with keyframe extraction to enable preference-aware optimization. Li et al. (2025) generate contrastive response pairs based on query-frame relevance, then jointly apply SFT and Direct Preference Optimization (DPO) to align the MLLM's outputs with human-like reasoning patterns. Inspired by the idea of Group Relative Policy Optimization (GRPO) algorithm, TSPO (Tang et al., 2025a) constructs a minimal-parameter Temporal Agent trained selectively on two challenging benchmarks (comprehensive temporal understanding and Needle-in-a-Haystack), demonstrating improved long-video reasoning through lightweight policy adaptation.

**Training-free Keyframe Extraction.** Training-free approaches can be broadly categorized into three paradigms based on their optimization logic: Pre-processing, Built-in Compression and Iterative Refinement. Pre-processing methods aim to deliver a curated set of keyframes to the MLLM prior to inference: VideoTree (Wang et al., 2025b) organizes long videos into hierarchical tree structures via clustering, enabling top-down

keyframe search with spatial-temporal coherence. CoS (Hu et al., 2025a) first encodes video stream into binary coding to perform pseudo temporal grounding, then feeds them into the MLLM for co-reasoning. BOLT (Liu et al., 2025a) and AKS (Tang et al., 2025b) respectively identify the cumulative distribution and local peaks along the CLIP-based relevance score curve to efficiently extract salient frames without exhaustive scanning, while Ye et al. (2025) perform multi-round temporal search using keyword-driven object detection, dynamically updating the relevance distribution of the frame sequence. Nar-KFC (Fang et al., 2025) constructs interleaved image-text streams that preserve both semantic relevance and temporal continuity, enhancing the MLLM's ability to track evolving events.

In contrast, Built-in Compression methods focus on increasing the token capacity and efficiency of MLLMs during inference. SF-LLaVA (Xu et al., 2024) introduces parallel token transmission channels with varying sampling rates and pooling intensities, achieving a free lunch on performance improvement. AdaRETAKE (Wang et al., 2025a) performs adaptive token compression over both the temporal dimension and transformer layers, dynamically allocating compression ratios to enable efficient processing of thousands of frames in a single pass. APVR (Gao et al., 2025) unifies query-aware semantic expansion with adaptive visual token selection, performing hierarchical key information extraction at both frame and token levels.

Iterative Refinement methods exploit the MLLM's intrinsic reasoning and self-reflection capabilities to dynamically adjust the keyframe set during inference. VideoAgent (Wang et al., 2024), as an early pioneer, iteratively selects frames based on the LLM's confidence level over current answers and associated captions. Ma et al. (2025) reformulates long-video understanding as a long-document retrieval task, employing multi-stage agent interaction to progressively refine the quality of retrieved contents. E-VRAG (Xu et al., 2025) explores multi-round self-reflection mechanisms within MLLMs, combined with hierarchical filtering of video content, to achieve effective long-video comprehension with relatively low computational overhead.

## 3 Methods

In this section, we will provide a detailed explanation of our **MIRA**[1] framework. We begin with introducing three different yet related frame scoring metrics (§3.1-§3.3 ) that form the foundation for our multi-view frame retrieval. Next, we combine these metrics to select keyframes that consider both relevance and causality (§3.4). Finally, we present the concept of a model feedback loop, designed to adapt the system to various user queries (§3.5).

### 3.1 Global-Regional Query-Frame Relevance Scoring

Following previous works (Tang et al., 2025b; Liu et al., 2025a), we first utilize CLIP to perform frame-by-frame feature extraction for the given video $\mathbf{V}_{1:T} \in \mathbb{R}^{T \times C \times H \times W}$ ($T, C, H, W$ are the number of frames and channels, height and width, respectively). $\mathbf{F}_{1:T} = \mathrm{CLIP}_\mathrm{I}(\mathbf{V}_{1:T}) \in \mathbb{R}^{T \times D}$ denotes the visual features processed by the CLIP vision tower with dimension $D$.

For a given user query, we compute the query-frame relevance score based on these visual features. While global-level semantics capture the overall scene, regional-level semantics are crucial for matching specific keywords in the user query. We capture these patch-level features as $\hat{\mathbf{F}}_{1:T} = \mathrm{CLIP}_\mathrm{I}(\mathbf{V}_{1:T,1:p^2}) \in \mathbb{R}^{T \times p^2 \times D}$ ($p^2$ is the number of visual patches). Therefore, to achieve more discriminative scoring, we propose a hybrid scoring that integrates both global and regional relevance.

$$\boldsymbol{S}_\mathrm{R}(\mathbf{V}_t, \mathbf{Q}) = \alpha \cdot \frac{\mathbf{F}_t \cdot \mathbf{Q}}{\|\mathbf{F}_t\|\|\mathbf{Q}\|} + (1 - \alpha) \sum_i^{p^2} \frac{\hat{\mathbf{F}}_{t,i} \cdot \mathbf{Q}}{\|\hat{\mathbf{F}}_{t,i}\|\|\mathbf{Q}\|}. \tag{1}$$

Here $\mathbf{Q} \in \mathbb{R}^{1 \times D}$ is the textual feature processed by the CLIP text tower and $\alpha \in [0, 1]$ controls the relative importance between global and local relevance.

---

[1]Project URLs: https://github.com/hzc1208/MIRA

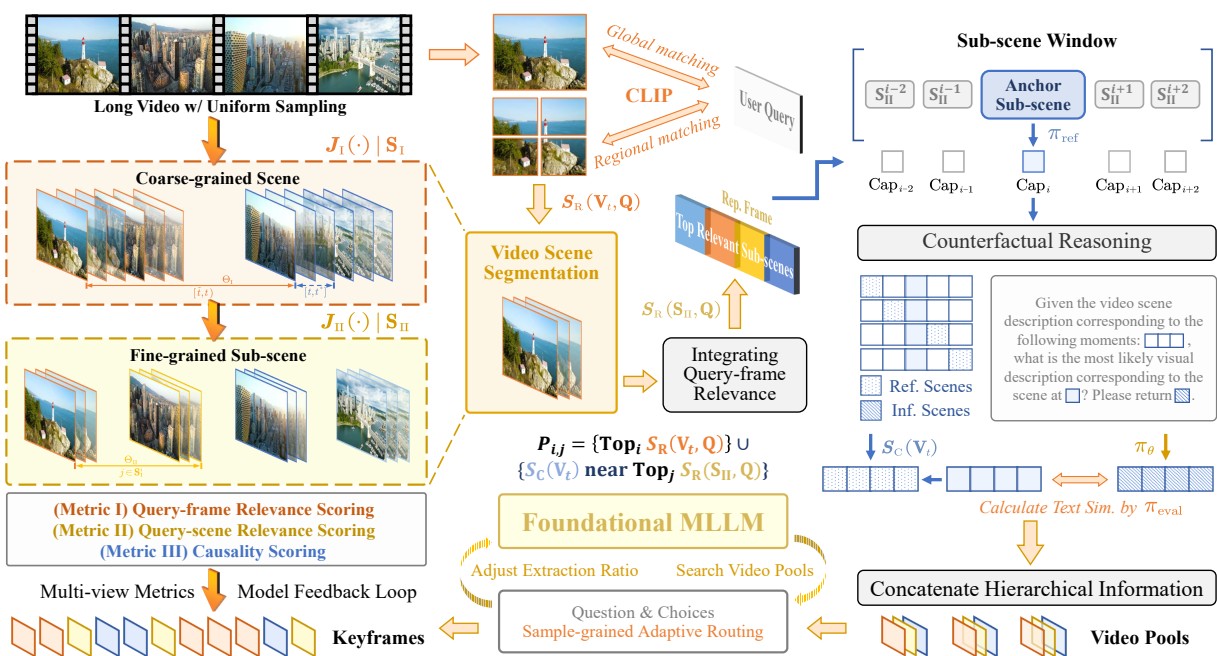

Figure 2: An overview of **MIRA** framework: for the given video and user query, we sequentially perform hierarchical scene construction (left), joint relevance calculation (mid-top), representative frame and sub-scene window modeling (right-top), causality evaluation (right-bottom) and closed-loop video pool selection (mid-bottom).

## 3.2 Hierarchical Scene Segmentation For Query-Scene Relevance Scoring

Solely relying on query-frame relevance may ignore the broader event context, leading to a fragmented understanding of the video's content. Therefore, we develop the second metric based on scene segments.

**Coarse-grained Scene Segmentation.** Scene boundaries are identified by detecting significant shifts in the features of adjacent frames. For a given scene starting index $\tilde{t}$, we define a judgment function $J_{\mathrm{I}}(\cdot)$, which is used to confirm whether the index interval $[\tilde{t}, t]$ will be split into an independent video scene: $J_{\mathrm{I}}(\tilde{t}, t) := \left( \max_{i \in [\tilde{t}, t), \, j \in [t, t^*]} \frac{\mathbf{F}_i \cdot \mathbf{F}_j}{\|\mathbf{F}_i\| \|\mathbf{F}_j\|} < \Theta_{\mathrm{I}} \right)$. Notably, to improve robustness against transient shot changes within a coherent scene, our judgment function incorporates a look-behind window. Specifically, when evaluating the current frame $\mathbf{V}_t$, we look back to a posterior frame $\mathbf{V}_{t^*}$. A new scene boundary is suppressed if any frame within the interval $(t, t^*]$ maintains semantic coherence with the established scene. This prevents the model from prematurely creating a new scene due to a brief cutaway, even when the feature similarity at frame $\mathbf{V}_t$ itself is low. $\Theta_{\mathrm{I}}$ represents cosine similarity threshold. The resulting scenes are denoted as $\mathbf{S}_{\mathrm{I}}$.

**Fine-grained Sub-scene Segmentation.** While coarse-grained segmentation identifies major semantic shifts, the resulting scenes are often too broad. They can simultaneously overlook subtle yet important visual changes while also containing significant temporal redundancy. To address both issues, we perform a second, finer-grained judgment within each scene: $J_{\mathrm{II}}(i, t) := \left( t \in \arg_k \min_{j \in \mathbf{S}_{\mathrm{I}}^i} \frac{\mathbf{F}_j \cdot \mathbf{F}_{j+1}}{\|\mathbf{F}_j\| \|\mathbf{F}_{j+1}\|} \ \wedge \ \frac{\mathbf{F}_t \cdot \mathbf{F}_{t+1}}{\|\mathbf{F}_t\| \|\mathbf{F}_{t+1}\|} < \Theta_{\mathrm{II}} \right)$.

Specifically, we segment the current scene $\mathbf{S}_{\mathrm{I}}^i$ for $k$ times based on the expected average sub-scene length, with the splitting positions being the indices corresponding to the bottom-$k$ similarities between adjacent frames, here $J_{\mathrm{II}}(i, t)$ represents whether sub-scene segmentation will be performed between the $t$-th and $(t+1)$-th frames in $\mathbf{S}_{\mathrm{I}}^i$. In addition, for positions with minimal visual changes (*i.e.* $\frac{\mathbf{F}_t \cdot \mathbf{F}_{t+1}}{\|\mathbf{F}_t\| \|\mathbf{F}_{t+1}\|} \geq \Theta_{\mathrm{II}}$), we will skip segmentation. Eventually, we obtain sub-scenes $\mathbf{S}_{\mathrm{II}}$ ready for query-scene relevance scoring.

**Query-Scene Relevance Scoring.** The relevance score for any sub-scene can be directly computed by summing the corresponding query-frame scores: $\boldsymbol{S}_{\mathrm{R}}(\mathbf{S}_{\mathrm{II}}^i, \mathbf{Q}) = \sum_{t \in \mathbf{S}_{\mathrm{II}}^i} \boldsymbol{S}_{\mathrm{R}}(\mathbf{V}_t, \mathbf{Q})$. Since the frames within the same sub-scene have high similarity, we consider extracting one **representative frame** for each sub-scene, which not only effectively reduces temporal redundancy, but also facilitates efficient processing in subsequent steps.

$$\mathbf{V}_{\mathrm{Rep}}^i = \arg \max_{\mathbf{V}_t \in \mathbf{S}_{\mathrm{II}}^i} \boldsymbol{S}_{\mathrm{R}}(\mathbf{V}_t, \mathbf{Q}). \tag{2}$$

Compared to query-frame relevance in the previous section, query-scene relevance scoring balances the high relevance of specific, peak moments with the broader semantic context provided by the entire scene.

### 3.3 Group-based Causal Modeling and Counterfactual Influence Scoring

The relevance scoring methods detailed in the previous sections are effective at identifying what frames and scenes are visually related to a user's query. However, relevance alone is often insufficient for a comprehensive understanding, as it fails to distinguish between the peak moment of an event and the crucial preceding moments that explain why or how it occurred. For example, a high relevance score might identify a building collapsing, but true comprehension requires identifying the initial explosion that caused it. To address this gap, this section introduces a third, complementary scoring metric derived from Group-based Causal Modeling. The resulting Counterfactual Influence Score is designed specifically to identify frames that provide this essential explanatory and causal context, moving beyond simple relevance to capture the video's underlying narrative structure.

**Constructing Sub-scene Windows.** Based on the query-scene scoring, we extract top relevant sub-scenes $\arg_k \max_{\mathbf{S}_{\mathrm{II}}} \boldsymbol{S}_{\mathrm{R}}(\mathbf{S}_{\mathrm{II}}, \mathbf{Q})$ and utilize them as the anchor point to expand forward and backward in time, thereby obtaining sub-scene windows composed of the corresponding representative frames. Now consider one single window constructed in this way. We denote the window by $\boldsymbol{w}|\boldsymbol{w}_{(i,2k+1)} = \{\mathbf{V}_{\mathrm{Rep}}^{i-k}, \cdots, \mathbf{V}_{\mathrm{Rep}}^i, \cdots, \mathbf{V}_{\mathrm{Rep}}^{i+k}\}$ and the anchor scene by $\boldsymbol{w}_c|\boldsymbol{w}_{(i,2k+1)} = \mathbf{V}_{\mathrm{Rep}}^i$. Here $\boldsymbol{w}_n = \boldsymbol{w} \setminus \{\boldsymbol{w}_c\}$ represents the contextual set composed of neighboring sub-scenes within the window. Our goal is to identify which sub-scenes in $\boldsymbol{w}_n$ have stronger causal relationships with $\boldsymbol{w}_c$, which can better facilitate MLLM's understanding of visual content in keyframes.

**Causality Evaluation.** We assess the causal influence of sub-scenes within a context window in an interactive scenario, which consists of an inference model $\pi_\theta$, a reference model $\pi_{\mathrm{ref}}$ and an evaluation model $\pi_{\mathrm{eval}}$. Considering that text is more conducive than visual signal to completing inference tasks, we respectively utilize a LLM, a Captioner model and $\mathrm{CLIP}_{\mathrm{T}}$ as $\pi_\theta, \pi_{\mathrm{ref}}, \pi_{\mathrm{eval}}$, thus projecting the visual information into the textual space. Guided by the principle of counterfactual intervention, we quantify each sub-scene's causal influence on its neighbors by masking them in turn. Concretely, we first use $\pi_{\mathrm{ref}}$ to generate captions for representative frames in each sub-scene window and request that the captions should highlight content related to the user query. For each $(2k+1)$-window $\boldsymbol{w}$, we can construct a set of examples with batch-size equal to $2k$: $\{\mathbf{V}_i : \boldsymbol{w}_n \setminus \{\mathbf{V}_i\}, \forall \mathbf{V}_i \in \boldsymbol{w}_n\}$. Here we select a frame $\mathbf{V}_i$ to be evaluated each time and pass the caption set corresponding to $\boldsymbol{w}_n \setminus \{\mathbf{V}_i\}$ into $\pi_\theta$ to infer the visual description of $\boldsymbol{w}_c$. Then, we compare the inferred caption with the actual caption generated by $\pi_{\mathrm{ref}}$ before. If there is a significant difference, it indicates a strong causal dependence between $\mathbf{V}_i$ and $\boldsymbol{w}_c$ and we need to refer to $\mathbf{V}_i$ to better understand $\boldsymbol{w}_c$. To this end, the Counterfactual Influence Score for each selected frame in the window is calculated as follows:

$$\begin{aligned}
\boldsymbol{S}_{\mathrm{C}}(\mathbf{V}_t) &= \sum_{\boldsymbol{w}|\mathbf{V}_t \in \boldsymbol{w}_n} \pi_{\mathrm{eval}}\left(\pi_\theta, \boldsymbol{w}_c | \pi_{\mathrm{ref}}, \boldsymbol{w}, \mathbf{V}_t\right) \\
&= \sum_{\boldsymbol{w}|\mathbf{V}_t \in \boldsymbol{w}_n} \sqrt{1 - \left(\frac{\tilde{\mathbf{F}}_{\pi_\theta(\boldsymbol{w}_c|\pi_{\mathrm{ref}}(\boldsymbol{w}_n \setminus \{\mathbf{V}_t\}))} \cdot \tilde{\mathbf{F}}_{\pi_{\mathrm{ref}}(\boldsymbol{w}_c)}}{\|\tilde{\mathbf{F}}_{\pi_\theta(\boldsymbol{w}_c|\pi_{\mathrm{ref}}(\boldsymbol{w}_n \setminus \{\mathbf{V}_t\}))}\| \|\tilde{\mathbf{F}}_{\pi_{\mathrm{ref}}(\boldsymbol{w}_c)}\|}\right)^2}.
\end{aligned} \tag{3}$$

For a given frame $\mathbf{V}_t$, we integrate the importance levels of $\mathbf{V}_t$ in all its relevant windows $\{\boldsymbol{w}|\mathbf{V}_t \in \boldsymbol{w}_n\}$. $\tilde{\mathbf{F}}_{\pi_\theta(\boldsymbol{w}_c|\pi_{\mathrm{ref}}(\boldsymbol{w}_n \setminus \{\mathbf{V}_t\}))}$ and $\tilde{\mathbf{F}}_{\pi_{\mathrm{ref}}(\boldsymbol{w}_c)}$ denote the textual features processed by $\pi_{\mathrm{eval}}$ for $\pi_\theta$'s and $\pi_{\mathrm{ref}}$'s captions, respectively.

### 3.4 The Construction of Video Keyframe Sampling Pools

We construct the final keyframe set by integrating frames from the above scoring metrics. Concretely, Query-Frame Relevance Score ($S_R(\mathbf{V})$) targets locally continuous moments of high correlation with the query, Query-Scene Relevance Score ($S_R(\mathbf{S}_{II})$) identifies the most crucial summary content, while Counterfactual Influence Score ($S_C$) highlights frames that provide essential explanatory context. A strategic combination of these three types of frames yields a comprehensive and robust keyframe selection for the video. Here we discuss two combining schemes.

**Scheme 1: Global Causality.** A straightforward selection strategy is to directly combine frames of high relevance and strong causality, as formulated in Eq.(4). It is worth noting that all the causal frames are drawn from context windows of highly correlated sub-scenes, hence the keyframe set implicitly encodes their representative content, even without explicitly selecting frames from these sub-scenes.

$$\mathbf{P}_{i:j} = \arg_i \max_{\mathbf{V}_t | t \in [1,T]} S_R(\mathbf{V}_t, \mathbf{Q}) \ \cup \ \arg_j \max_{\mathbf{V}_t \in \{\boldsymbol{w}_n\}} S_C(\mathbf{V}_t). \tag{4}$$

**Scheme 2: Intra-group Causality.** Another approach is to incorporate both the representative frame and the causal frame in every high-valued sub-scene. Therefore, out of each top-ranking sub-scene window that has high query-scene relevance score we select two frames: the representative frame at the central-anchor position and the causal frame that has the highest Counterfactual Influence Score within the context window. Eq.(5) represents the entire construction process.

$$\mathbf{P}_{i:j} = \arg_i \max_{\mathbf{V}_t | t \in [1,T]} S_R(\mathbf{V}_t, \mathbf{Q}) \ \cup \ \left\{ \boldsymbol{w}_c, \arg \max_{\mathbf{V}_t \in \boldsymbol{w}_n} S_C(\mathbf{V}_t) | \boldsymbol{w} \in \arg_j \max_{\boldsymbol{w}} S_R(\boldsymbol{w}_c, \mathbf{Q}) \right\}. \tag{5}$$

**Evaluating Performance Diversity and Upper Bound in the Video Sampling Pool.** It is interesting to analyze how different strategies in the pool affect the model response. Figure 3 visualizes the response behavior of seven different sampling strategies in the video pool across LVC benchmarks. The horizontal axis indicates the proportion of sampling strategies that correctly answer each sample, while the grid entries denote the number of samples correctly answered by a specific strategy. As we move rightward along the axis, query difficulty increases, and the intersection of correct responses correspondingly shrinks. For instance, Column 7/7 corresponds to samples that all seven sampling strategies lead to the correct answer. In contrast, Column 1/7 contains samples that only a single particular sampling strategy yields the correct answer. By comparing the numbers in column 1/7, we can conclude that the existence of those extreme sampling strategies in pool, such as pool-1 and pool-7 hold the value of increasing the diversity of responses and enhancing the potential of the pool to handle difficult queries. In the ideal condition, if one could perfectly match each sample to the strategy that yields correct answer, the resulting performance would represent the upper-bound achievable by combining all strategies in the pool, which has denoted as Upper Acc in Fig.3.

The gap between the theoretical upper-bound and the actual best performance for any single sampling strategy demonstrates that even with strategic frame selection method, the long video sampling bias issue persists: MLLMs only process a sparse slice of frames eventually, each different sampling highlights a different "story", causing the model to generate distinct answers to the same query. The existence of such sampling bias reveals the shortcoming of the conventional open-loop frame selection pipeline for long video understanding: no room for the adjustment of the selected frames. In light of this, we further propose a closed-loop solution by integrating MLLM's feedback to the inference pipeline.

### 3.5 Adaptive Routing Strategy based on Model Feedback Loop

We introduce two types of feedback. The first allows the model to analyze the user query and report the desired sampling before receiving video frames, while the second allows the model to say "I don't know" after looking at the bias-sampled video frames and requires more information. In other words, MLLMs guide the frame selection process in our closed-loop pipeline as an agent. We explain the details in the following.

**Closed-loop pipeline 1: User Query Analysis.** This pipeline considers the inherent heterogeneity of user queries in the domain of LVC, ask the MLLM to propose the desired video pool prior to the sampling

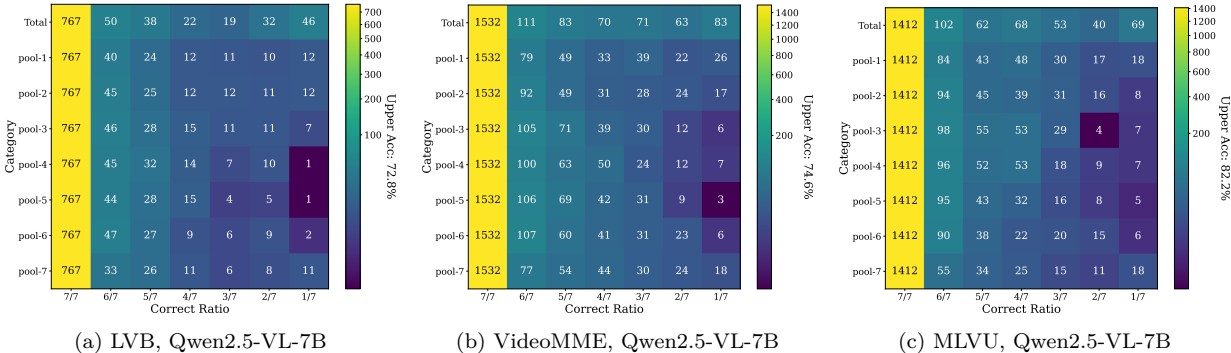

(a) LVB, Qwen2.5-VL-7B  (b) VideoMME, Qwen2.5-VL-7B  (c) MLVU, Qwen2.5-VL-7B

Figure 3: Distributions of correctly answered samples in multiple video sampling pools. Here the $j$-th column of pool-$i$ is the number of samples answered correctly by $j$ video pools (including pool-$i$) simultaneously; **Upper Acc.** denotes the accuracy upper-bound achievable across all video pools, wherein a sample is deemed correct provided that any individual pool yields a correct answer.

process, and provides the corresponding frames adaptively. The diversity of our video pool allows the query-based routing to end up with frames suitable to answer the given query. Before providing the MLLM with sampled video frames, we let the model first parse and classify the user queries. The model is allowed to determine if looking at only a single scene is sufficient to answer the query, or multiple scenes are necessary to understand the context. Different parsing results of the user query lead to different candidates in our video pool.

**Closed-loop pipeline 2: Option of Reporting Missing Information.** This pipeline allows the MLLM to judge whether it received sufficient information from the sampled video frames to answer a particular query. The model is provided with an additional option to refuse answering the query whenever it finds the sampled video frames are not enough for determine the answer. Using multiple-choice questions as an example, we expand the option list to include the choice of "Insufficient visual information". Whenever the model chooses this option as an answer, we will increase the capacity of the video pool and repeat the question. This pipeline guarantees that the model's final answer to the question is a fully-informed response.

## 4 Experiments

To validate the effectiveness of our framework, we choose LLaVA-Video (Zhang et al., 2024b) and Qwen2.5-VL (Bai et al., 2025) as our baseline models, then evaluate on three widely adopted benchmarks in the domain of video understanding: LVB (Wu et al., 2024), VideoMME (Fu et al., 2025) and MLVU (Zhou et al., 2024).

**LongVideoBench (LVB)** contains videos ranging from 8 seconds to 60 minutes across diverse topics, accompanied by highly detailed user queries (average length: 43.53 words). We choose its publicly available validation set, comprising 1,337 QA pairs grounded in 752 videos.

**VideoMME** includes 2,700 QA pairs with an average video duration of 1,017.9 seconds. Its queries are notably concise (average length: 35.7 tokens), posing a significant LVC challenge without leveraging external subtitles.

**MLVU** features videos up to 2 hours in length and covers multiple complex task types, including Holistic, Single-Detail and Multi-Detail LVC. We select its multiple-choice subset, consisting of 2,174 QA pairs.

Following prior works, we employ LMMs-Eval (Zhang et al., 2024a) as the evaluation toolkit. For LVB and VideoMME, we report overall accuracy; for MLVU, we report average accuracy across its multi-task setup. Unless otherwise specified, our framework extracts a fixed total of 64 keyframes per video. As defined in Eqs.(4)-(5), we generate video pools with varying blending ratios by setting $i \in \{8n | n \in [0,8] \land n \in \mathbb{N}\}$. Specifically, for each sample, we first select the top-$i$ frames from the perspective of relevance. Then we

Table 1: Performance comparison among foundational (proprietary & open-source) MLLMs, training-based & free methods and our framework on three challenging LVC benchmarks. Here the numbers in parentheses indicate the performance improvement compared to the corresponding baseline model after introducing our framework, FPS denotes frames per second.

| Method | Type | Model | Frames | LVB | VideoMME | MLVU |
|---|---|---|---|---|---|---|
| GPT-4o mini | Foundational | - | - | 56.5 | 64.8 | - |
| GPT-4V | Foundational | - | 384 | 60.7 | 59.9 | 49.2 |
| GPT-4o (Hurst et al., 2024) | Foundational | - | 384 | 66.7 | 71.9 | 64.6 |
| LLaVA-OV (Li et al., 2024) | Foundational | LLaVA-OV-7B | 32 | 56.5 | 58.2 | 64.7 |
| NVILA (Liu et al., 2025b) | Foundational | NVILA-8B | 256 | 57.7 | 64.2 | 70.1 |
| LLaVA-Video (Zhang et al., 2024b) | Foundational | LLaVA-Video-7B | 64 | 58.2 | 63.3 | 70.8 |
| Qwen2.5-VL (Bai et al., 2025) | Foundational | Qwen2.5-VL-7B$^{\dagger}$ | 64 | 60.0 | 63.5 | 63.2 |
| LongVILA (Chen et al., 2024a) | Training-based | LongVILA-7B | 256 | 57.1 | 60.1 | - |
| LongVU (Shen et al., 2024) | Training-based | LongVU-7B | 1FPS | - | 60.6 | 65.4 |
| Apollo (Zohar et al., 2025) | Training-based | Apollo-7B | 2FPS | 58.5 | 61.3 | 70.9 |
| BIMBA (Islam et al., 2025) | Training-based | BIMBA-7B | 128 | 59.5 | 64.7 | 71.4 |
| TPO (Li et al., 2025) | Training-based | LLaVA-Video-7B | 64 | 60.1 | 65.6 | 71.1 |
| BOLT (Liu et al., 2025a) | Training-free | LLaVA-OV-7B | 32 | 59.6 | 59.9 | 66.8 |
| CoS (Hu et al., 2025a) | Training-free | LLaVA-Video-7B | 64 | 58.9 | 64.4 | 71.4 |
| AKS (Tang et al., 2025b) | Training-free | LLaVA-Video-7B | 64 | 62.7 | 65.3 | - |
| | | Qwen2.5-VL-7B$^{\dagger}$ | | 61.3 | 64.8 | - |
| **MIRA** | Training-free | LLaVA-Video-7B | 64 | **64.5** (+6.3) | **66.2** (+2.9) | **73.8** (+3.0) |
| | | Qwen2.5-VL-7B | | **66.9** (+6.9) | **67.0** (+3.5) | **76.3** (+13.1) |

$^{\dagger}$ denotes our reproduced experimental results.

sequentially augment the selection with frames chosen according to summarization or causality-oriented criteria, following the designated algorithm mentioned above. If the total frame count remains below 64 after this process, which indicates limited sub-scene diversity in the video, we supplement the selection by greedily choosing additional frames from the unselected frame set, ranked by descending relevance score, until the 64-frame capacity is reached. Specific experimental details and algorithm procedure of **MIRA** are provided in §A.

## 4.1 Main Results

We systematically categorize existing LVC works into three groups: (i) Foundational models, further divided into proprietary and open-source variants; (ii) Training-based models; (iii) Training-free methods. For MLLMs lacking keyframe selection component, we apply uniform sampling for the given video by default. Table 1 presents a comprehensive performance comparison among these categories and our framework.

First, compared to the respective baseline models, our method achieves consistent and significant gains: +2.9% to +6.3% on LLaVA-Video-7B and +3.5% to +13.1% on Qwen-2.5-VL-7B across all benchmarks. Second, as a plug-and-play scheme, **MIRA** remains competitive even against training-based models. For instance, under the condition of utilizing LLaVA-Video-7B as the baseline model, we outperform TPO (Li et al., 2025) by 4.4% on LVB and 2.7% on MLVU. Third, when compared with similar training-free methods, **MIRA** also demonstrates clear superiority: on the competitive Qwen2.5-VL-7B model, it surpasses AKS (Tang et al., 2025b) by 5.6%, 2.2% on LVB and VideoMME, respectively. In addition, it is worth noting that our approach even exceeds the performance of all proprietary models on LVB and MLVU, including GPT-4o (Hurst et al., 2024), which highlights the potential of the keyframe retrieval paradigm proposed in this work.

Table 2: Comparison of open-loop and closed-loop pipelines on three benchmarks (64 frames).

| Sampling Strategy | Qwen2.5-VL-7B | | | LLaVA-Video-7B | | |
|---|---|---|---|---|---|---|
| | LVB | VideoMME | MLVU | LVB | VideoMME | MLVU |
| Uniform Sampling | 60.0 | 63.5 | 63.2 | 58.2 | 63.3 | 70.8 |
| Fixed Sampling | 66.2 | 66.7 | 75.5 | 63.9 | 65.9 | 73.5 |
| User Query Analysis | **66.9** | 66.9 | 75.9 | 63.9 | **66.2** | **73.8** |
| User Query Analysis† | 66.7 | **67.0** | **76.3** | **64.5** | 66.0 | **73.8** |
| Missing Information | 66.3 | 66.8 | 75.4 | 63.7 | 65.4 | 73.7 |

† denotes utilizing a LLM to assist in analyzing the given query.

Table 3: Ablation studies for different causal frame selection schemes described in §3.4.

| Selection Scheme | Qwen2.5-VL-7B | | | LLaVA-Video-7B | | |
|---|---|---|---|---|---|---|
| | LVB | VideoMME | MLVU | LVB | VideoMME | MLVU |
| Global Causality | 65.1 | 65.8 | 73.8 | **63.4** | 65.1 | 72.8 |
| Intra-group Causality | **66.2** | **66.5** | **75.5** | 62.2 | **65.5** | **73.0** |

## 4.2 Comparison of Open-loop and Closed-loop Pipelines

As shown in Tab.2, we compare the performance of different sampling strategies. Here Fixed Sampling denotes using an optimal and fixed blending ratio when selecting keyframes for all user queries, which is a conventional open-loop pipeline for inference. In comparison, User Query Analysis and Missing Information refer to the two closed-loop pipelines that leverage feedback from the model to adjust the sampling strategy (§3.4). In Query Analysis, the sampling strategy alternates between the two best candidates in the video pool depending on whether the model determines to focus more on the highest-relevant scenes or the causal content of related events. In Missing Information, models are first given 64 sampled video frames using the same blending ratio as Fixed Sampling. If the model chooses the additional option of insufficient information, we increase the frame number to 128 on those test samples (around 15%) for more visual cues, and we run the inference on them for the second time. No additional option will be given in the second round.

Experimental results suggest that Query Analysis strategies can further improve upon this baseline by up to 0.7%-0.8%, demonstrating the superiority of introducing a dynamic adjustable feedback loop rather than relying on a fixed, one-size-fits-all pool. However, no consistent improvement is observed using the Missing Information strategy. Through an in-depth analysis, we discover two reasons. First, this feedback is unreliable, as current MLLMs may not be able to effectively determine whether the visual information is sufficient. Second, adding an alterative option decreases the performance in general. Nevertheless, the performance improvement in partial cases indicates that powerful MLLMs have the potential of understanding long videos as an agent.

## 4.3 Ablation Studies for Different Types of Keyframes

**Causal frame selection scheme.** Table 3 makes a comparison of the two causal frame extraction schemes mentioned in Eqs.(4)-(5), here $i : j = 40 : 24$. Compared to simply extracting a subset of frames with the strongest causality, comprehensive extraction of summary and causal frames based on sub-scene windows generally achieves superior performance, indicating that causal frames may be more suitable for playing the role of auxiliary understanding in LVC tasks.

**The effectiveness of each component.** In Tab.4, we investigate how keyframe composition, which is derived from different retrieval criteria, affects overall LVC performance within a single video pool. The configuration *a.* denotes a greedy selection based solely on relevance scores, without sub-scene window modeling or causality-aware evaluation, *i.e.* Fig.2(II). In contrast, configuration *b.* and *c.* introduce a moderate number of frames selected from the perspectives of summarization or causality, respectively. *d.*

Table 4: Ablation studies for mixing keyframes from multiple perspectives. Here *Rel., Sum., Causal.* denotes a specific video pool composed of 64 frames extracted from the perspectives of relevance, summarization and causality, respectively.

| Video Pool Configuration | | | Qwen2.5-VL-7B | | | LLaVA-Video-7B | | |
|---|---|---|---|---|---|---|---|---|
| Rel. | Sum. | Causal. | LVB | VideoMME | MLVU | LVB | VideoMME | MLVU |
| Uniform Sampling | | | 60.0 | 63.5 | 63.2 | 58.2 | 63.3 | 70.8 |
| *a.* ✓ | - | - | 65.3 | 64.8 | 74.4 | **63.9** | 64.4 | 73.2 |
| *b.* ✓ | ✓ | - | 65.7 | 65.4 | 74.4 | 62.8 | **65.5** | **73.6** |
| *c.* ✓ | - | ✓ | 65.4 | 66.3 | 74.9 | 62.7 | 64.9 | 72.9 |
| *d.* ✓ | ✓ | ✓ | **66.2** | **66.5** | **75.5** | 62.2 | **65.5** | 73.0 |

represents extracting keyframes according to the second scheme as in Eq.(5). Here we set the total frame number of $j$ to 24. Experimental results demonstrate that selecting keyframes based on relevance alone does not yield optimal performance in most cases. Instead, the effective combination of all three keyframe types is essential to fully unleash the inherent reasoning capability of MLLMs. Results with the feedback loop can be found in Tab.2 and are thus not included here.

# 5 Conclusions

In this work, we present a test-time video comprehension framework that jointly extracts keyframes from three complementary perspectives: relevance, summarization, and causality. We demonstrate that blending these multi-view cues at varying ratios allows the model to better adapt to user queries with diverse intents. To this end, we propose multiple closed-loop strategies that dynamically assign an optimal blending ratio or keyframe capacity to each sample, aiming to harness the full reasoning potential of MLLMs. We believe that future research in LVC should focus on developing more precise control mechanisms for modulating multi-level information and enabling sample-specific allocation.

# Broader Impact Statement

MIRA framework has the properties of multi-stage, training-free and closed-loop, which can support LVC tasks in both online and offline scenarios. Calling the relevant model processing interfaces locally will further enhance the security of LVC.

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

# A  Appendix

## A.1  Experimental Configuration

**Hyper-parameter Settings.** Prior to hierarchical scene construction, we perform uniform frame sampling for the given video, as illustrated in Fig.2(I). For all experiments reported in this work, we adopt a default sampling rate of 1 FPS. For videos shorter than 128 seconds, we dynamically adjust the sampling interval to ensure that at least 128 frames are extracted, preserving sufficient temporal coverage for downstream processing.

In constructing first-level scenes and second-level sub-scenes, we set $t^* = \min(t + 2, T), \Theta_{\mathrm{I}} = \cos(\pi/4), \Theta_{\mathrm{II}} = \cos(\pi/12)$ and segment sub-scenes with an average length of 4 frames. Specifically, for video scene $\mathbf{S}_{\mathrm{II}}^i$ containing $n$ frames, we apply the segmentation criterion defined in $\boldsymbol{J}_{\mathrm{II}}(\cdot)$, where the partitioning parameter $k = \lfloor n/4 \rfloor$.

Table S1: Detailed performance summary of video pools based on different blending ratios. Here *Rel-i, Sum-Causal-j* denotes a specific pool composed of $i$ and $j$ frames, which are respectively extracted from the frame- and sub-scene- levels, according to the video pool synthesis process mentioned above. Here **Bold** and Underline denote the optimal and suboptimal results.

| Video Pool Configuration | Qwen2.5-VL-7B | | | LLaVA-Video-7B | | |
|---|---|---|---|---|---|---|
| | LVB | VideoMME | MLVU | LVB | VideoMME | MLVU |
| Uniform Sampling | 60.0 | 63.5 | 63.2 | 58.2 | 63.3 | 70.8 |
| *Rel-64, Sum-Causal-0* | 65.3 | 64.8 | 74.4 | **63.9** | 64.4 | 73.2 |
| *Rel-56, Sum-Causal-8* | 65.5 | 65.9 | 75.1 | 62.4 | 64.4 | 73.4 |
| *Rel-48, Sum-Causal-16* | 66.1 | 65.7 | 74.9 | 63.2 | 65.7 | **73.5** |
| *Rel-40, Sum-Causal-24* | **66.2** | 66.5 | **75.5** | 62.2 | 65.5 | 73.0 |
| *Rel-32, Sum-Causal-32* | 65.5 | 66.2 | 75.0 | 63.3 | 65.7 | 72.9 |
| *Rel-24, Sum-Causal-40* | 64.6 | 66.4 | 72.9 | 63.2 | **65.9** | 72.9 |
| *Rel-16, Sum-Causal-48* | 64.8 | **66.7** | 72.6 | 63.4 | 65.7 | 72.2 |
| *Rel-8, Sum-Causal-56* | 64.5 | 65.9 | 71.0 | 62.5 | 65.6 | 72.0 |
| *Rel-0, Sum-Causal-64* | 61.8 | 65.6 | 68.6 | 61.8 | 65.2 | 71.2 |

Table S2: Category-wise performance of video pools based on different blending ratios on LongVideoBench. The performance are computed based on four duration group, 15s, 60s, 600s and 3600s.

| Video Pool Configuration | Qwen2.5-VL-7B | | | | LLaVA-Video-7B | | | |
|---|---|---|---|---|---|---|---|---|
| | 15s | 60s | 600s | 3600s | 15s | 60s | 600s | 3600s |
| *Rel-64, Sum-Causal-0* | 71.4 | 75.6 | 65.0 | 60.3 | 68.8 | 73.3 | 63.1 | 59.9 |
| *Rel-56, Sum-Causal-8* | 74.6 | 76.7 | 64.6 | 59.8 | 67.2 | 72.1 | 61.7 | 58.3 |
| *Rel-48, Sum-Causal-16* | 74.6 | 76.7 | 65.8 | 60.3 | 68.8 | 72.1 | 62.4 | 59.2 |
| *Rel-40, Sum-Causal-24* | 74.6 | 76.2 | 66.0 | 60.5 | 68.8 | 72.1 | 62.1 | 57.1 |
| *Rel-32, Sum-Causal-32* | 74.1 | 76.7 | 64.6 | 59.9 | 68.8 | 72.1 | 62.6 | 59.2 |
| *Rel-24, Sum-Causal-40* | 74.1 | 76.7 | 63.6 | 58.5 | 68.8 | 72.1 | 62.1 | 59.4 |
| *Rel-16, Sum-Causal-48* | 74.1 | 76.7 | 63.1 | 59.4 | 68.8 | 72.1 | 61.9 | 60.1 |
| *Rel-8, Sum-Causal-56* | 74.1 | 76.7 | 62.4 | 59.0 | 68.8 | 72.1 | 62.1 | 57.8 |
| *Rel-0, Sum-Causal-64* | 74.1 | 76.7 | 60.9 | 53.7 | 68.8 | 72.1 | 61.4 | 56.6 |

Sub-scene windows are initialized with a default length of 5 frames, truncated to a minimum of 4 frames when positioned at temporal boundaries. We retain up to 64 sub-scene windows as representative frames for summarization- and causality-based keyframe selection. Here we allow representative frames to overlap across windows. To avoid redundancy, we first deduplicate the aggregated representative frame set before feeding it into $\pi_{\text{ref}}$ to generate descriptive captions for each sub-scene. We then apply random masking over neighboring sub-scenes and pass the masked context windows to $\pi_\theta$ for causal reasoning.

**Employed Models.** We utlize FG-CLIP (Xie et al., 2025) for relevance calculation. Specifically, input frames are resized to $224 \times 224$ before being fed into the vision tower, while questions and corresponding choices are tokenized and truncated to a maximum of 248 tokens for the text tower. In Eq.(1), we set $\alpha = 0.5, D = 768, p = 24$.

For caption generation and causal inference, we utilize the lightweight Qwen2.5-VL and Qwen3 (Yang et al., 2025) models, respectively. For $\pi_{\text{ref}}$, we set max_pixels $= 128 \times 28 \times 28$, min_pixels $= 28 \times 28$ and constrain caption length to no more than 25 words generally. Both models are configured with max_model_len $= 8192$ and sampling parameters are performed with temperature $= 0$, top_p $= 0.95$, top_k $= 20$. All experiments are conducted on NVIDIA A100-SXM4-40GB and A800-SXM4-80GB GPUs.

Table S3: Category-wise performance of video pools based on different blending ratios on VideoMME. The performance are computed based on three duration group, short, medium and long.

| Video Pool Configuration | Qwen2.5-VL-7B | | | LLaVA-Video-7B | | |
|---|---|---|---|---|---|---|
| | Short | Medium | Long | Short | Medium | Long |
| *Rel-64, Sum-Causal-0* | 75.9 | 65.9 | 52.7 | 76.3 | 64.4 | 52.3 |
| *Rel-56, Sum-Causal-8* | 77.3 | 65.1 | 55.3 | 77.6 | 63.3 | 52.2 |
| *Rel-48, Sum-Causal-16* | 77.6 | 65.7 | 53.8 | 78.9 | 64.6 | 53.8 |
| *Rel-40, Sum-Causal-24* | 77.6 | 66.9 | 55.0 | 78.1 | 65.1 | 53.2 |
| *Rel-32, Sum-Causal-32* | 76.4 | 66.7 | 55.6 | 77.6 | 66.4 | 53.2 |
| *Rel-24, Sum-Causal-40* | 76.7 | 67.0 | 55.4 | 77.7 | 66.2 | 53.9 |
| *Rel-16, Sum-Causal-48* | 76.6 | 66.8 | 56.7 | 77.7 | 65.6 | 54.0 |
| *Rel-8, Sum-Causal-56* | 76.6 | 66.3 | 54.8 | 77.6 | 65.0 | 54.2 |
| *Rel-0, Sum-Causal-64* | 76.6 | 65.6 | 54.6 | 77.6 | 63.4 | 54.6 |

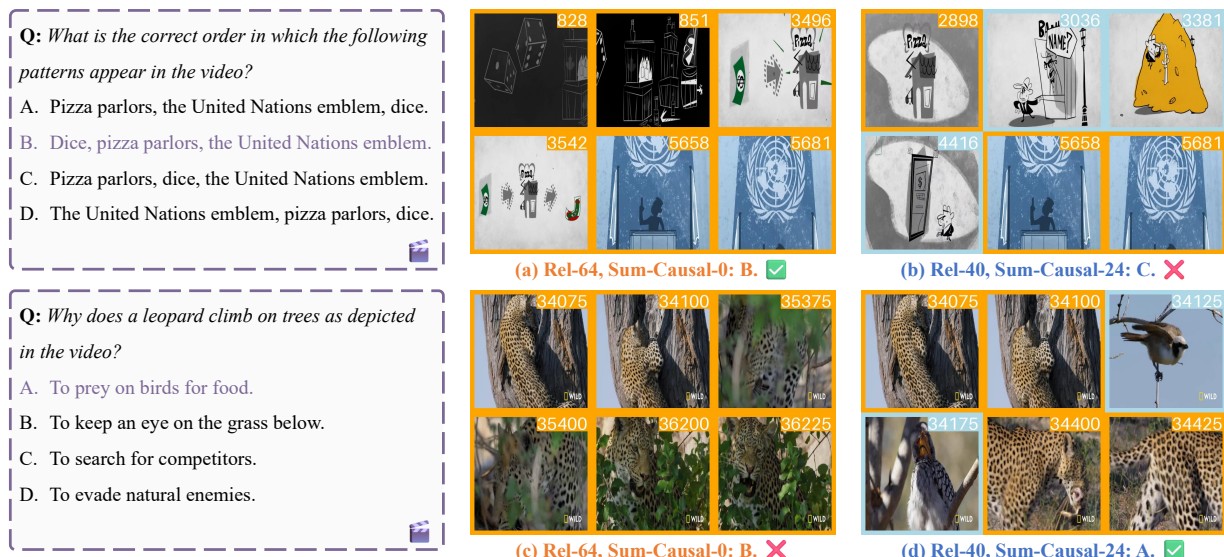

Figure S1: The response status of different video pools on Qwen2.5-VL-7B, VideoMME. Here *Rel-i, Sum-Causal-j* denotes a specific pool composed of *i* and *j* frames, which are respectively extracted from the frame- and sub-scene- levels, according to the video pool synthesis process mentioned above.

## A.2 Detailed Results on Three Benchmarks

This section expands upon the summary tables presented in the main paper by reporting full benchmark results across all three datasets. For each dataset, we provide comparisons with prior baselines as well as a breakdown of the performance variations across different video pools, as shown in Tabs.S1-S3.

## A.3 Visualization of Different Video Sampling Pools

Figure S1 illustrates the performance of video sampling pools with different mixture ratios in responding to user queries of varying intents. Pool *Rel-64, Sum-Causal-0*, which consists entirely of keyframes selected from a relevance-based perspective, is better suited for addressing local, detail-oriented questions, as exemplified by the case in the first row. In contrast, introducing a certain proportion of keyframes derived from the perspective of summarization and causal inference enables the MLLM to more effectively answer queries that require understanding of event context, such as the case shown in the second row.

### A.4 Algorithm

We provide the detailed algorithm procedure of **MIRA** in Alg.1. To summarize, the input video is first segmented into scenes and sub-scenes to establish a structured temporal hierarchy, enabling efficient management of long sequences. Then the frame-level and sub-scene-level relevance scores are computed with respect to the user query, providing an initial ranking of potentially informative segments. Highly relevant representative frames are extracted and used as anchors to form localized temporal windows, capturing context around critical events without processing the full sequence. Next, captions are generated for representative frames, while neighboring segments are masked to assess predictive consistency. This causality check selects the context frames that contain crucial information to understand the event in the video. Finally, multiple video pools are generated, each blending segments at varying ratios. A closed-loop routing strategy iteratively selects the most coherent path, yielding the final MLLM-generated responses aligned with query objectives.

---

**Algorithm 1 M**ulti-view **I**ntegrated **R**etrieval with **A**daptive Routing (**MIRA**)

---

**Input:** the given videos $\mathbf{V}_{(1:N),1:T}$ and user queries $\mathbf{Q}_{(1:N)}$, the inference, reference and evaluation model $\pi_\theta, \pi_{\text{ref}}, \pi_{\text{eval}}$ for causality modeling

**Output:** the MLLM response set $\tilde{\mathbf{A}}_{(1:N)}$

1: # *Hierarchical Video Scene Construction*
2: Apply $\boldsymbol{J}_{\text{I}}(\cdot), \boldsymbol{J}_{\text{II}}(\cdot)$ for $\mathbf{V}_{(1:N),1:T}$ to sequentially establish video scenes $\bigcup_{i=1}^{N} \mathbf{S}_{\text{I}}^{(i)}$ and sub-scenes $\bigcup_{i=1}^{N} \mathbf{S}_{\text{II}}^{(i)}$

3: # *Relevance Calculation*
4: Calculate the joint relevance scores $\bigcup_{i=1}^{N} \boldsymbol{S}_{\text{R}}(\mathbf{V}_{(i),1:T}, \mathbf{Q}_{(i)})$ frame by frame according to Eq.(1)
5: # *Sub-scene Window Modeling*
6: Calculate the relevance scores of video sub-scenes $\bigcup_{i=1}^{N} \boldsymbol{S}_{\text{R}}(\mathbf{S}_{\text{II}}^{(i)}, \mathbf{Q}_{(i)})$
7: Extract representative frames from $\bigcup_{i=1}^{N} \mathbf{S}_{\text{II}}^{(i)}$ based on Eq.(2)
8: Construct corresponding sub-scene windows $\bigcup_{i=1}^{N} \{\boldsymbol{w}_c, \boldsymbol{w}_n\}_{(i)}$ by utilizing top-$k$ relevant sub-scenes (representative frames) as central-anchor points
9: # *Causality Evaluation*
10: Generate captions for representative frames and randomly mask neighboring sub-scenes $\boldsymbol{w}_n$
11: Employ $\pi_\theta$ to predict the visual description at the central-anchor point corresponding to each set of mask data
12: Calculate the causal scores of relevant representative frames $\bigcup_{i=1}^{N} \boldsymbol{S}_{\text{C}}(\mathbf{V}_{(i),1:T} | \mathbf{V}_{(i),1:T} \in \{\boldsymbol{w}_n\}_{(i)})$ according to Eq.(3)
13: # *Closed-loop Routing Strategies*
14: Generate $M$ video pools $\mathbf{P}_{(1:N),(1:M)}$ with different blending ratios based on Eqs.(4)-(5)
15: Based on the existing video pools, choose a specific closed-loop pipeline to obtain the final model response set $\tilde{\mathbf{A}}_{(1:N)}$ of sample granularity

---

### A.5 Implementation Detail and Analysis of Closed-loop pipelines

In this final section we elaborate more on the implementation details and the analysis of the proposed closed-loop pipelines: User Query Analysis and Missing Information.

**User Question Analysis**. We adopt the following prompt to ask the MLLM classify the user query into "single" or "multiple" category: *Determine if the following question can be answered by viewing a single scene from the video, or if it requires understanding events and relationships across multiple scenes. Output \*exactly\* one lowercase word: "single" or "multiple". Do not include any other text, punctuation, or explanation. For example: Question: What color is the car? → single Question: What is the woman with the pink hat wearing → single Question: Why did the person run away? → multiple Question: What is the order of the following event? → multiple Question: < |placeholder| >.* The "< |placeholder| >" string will be replaced by the actual user query. No video frames are input into the model at this stage. After

the classification, we will determine the blending ratio of $i$ and $j$ for high-ranking frames and representative frames. For user queries that are identified as "single", we select more high-ranking frames, whereas the number of representative frames increase when the question is classified as "multiple".

**Missing Information**. We add an additional option in the candidate list of the user query. For instance, the original question and candidates are: *What is the color of the car? A. Red B. Gray C. White D. Black +* *(64 sampled video frames)* and the modified input becomes: *What is the color of the car? A. Red B. Gray C. White D. Black E. Insufficient visual information + (64 sampled video frames)*. If the model still responds with one of the original four options, we record its answer, and the inference ends. We call the union of these test samples as the sufficient set. If the answer is 'E', we sample more frames from the video and modify the input at a second time: *What is the color of the car? A. Red B. Gray C. White D. Black + (128 sampled video frames)*. These samples are included in the insufficient set. We decide to end at the second round for the simplicity of the pipeline. We have discovered interesting phenomenon when we analyzed the imperfect results of this pipeline, as shown in Tab.2. First of all, we observed that the new option influences the MLLM's decision even when the model thinks that the visual information provided is enough. As a result, the model performs worse on the sufficient set when given the first modified input. Therefore, even though increasing the sampling frames on the insufficient set improves the performance, the overall improvement is less significant. Second, by visualizing the samples identified as insufficient, we realized that the model cannot fully judge whether it receives enough visual clues from the video to answer the question. We hope that reinforcement learning on the model could help improve its capability of determining whether it is given a proper question. We will leave this as the future work.

