# OpenReview forum: "MIRA: Multi-view Information Retrieval with Adaptive Routing for Test-time Long-video Comprehension"
_TMLR — Accepted by TMLR_

### Review · Reviewer_LyCH · 2025-12-07

**Summary Of Contributions:**

The paper proposes MIRA, a multi-view information retrieval framework for long-video comprehension (LVC) with multimodal LLMs. MIRA combines three relevance scores (global + patch-level relevance score, a scene relevance score, and a caption-based causal influence score derived via counterfactual ablations) to generate multiple sampling pools that capture complementary “views” of the video, and a closed-loop model feedback mechanism to refine sampling when needed.

The authors evaluate MIRA on LVB, VideoMME, and MLVU benchmarks using two MLLM backbones, reporting consistent improvements over existing approaches.

The work aims to address the challenge that no single sampling strategy captures all aspects of long videos, and positions MIRA as a robust, multi-view alternative.


Strengths
-The paper clearly articulates the different approaches to LVC, and why single-strategy sampling is brittle.
-Methodological modularity: MIRA is training-free, model-agnostic, and easy to integrate with existing MLLMs.
-Causal-influence scoring is novel within LVC: Although related to counterfactual leave-one-out methods, the specific caption-based formulation is new in inference-time video sampling.
-Comprehensive benchmark coverage: Evaluation spans three LVC datasets and two MLLMs (LLaVA-Video (Zhang et al., 2024b) and Qwen2.5-VL (Bai et al., 2025)).

Weaknesses
1. The causal scoring metric risks missing temporally-distant causal events
2. Insufficient attribution of improvements to model features, with the majority of improvement from a single relevance score
3. Limited evaluation of causal claims of the causal relevance score
4. Missing key long-video benchmark: EgoSchema

Further information provided in the "Requested Changes" section

**Audience:**

Yes

**Audience Explanation:**

This work will be of interest to ML practitioners who work with video comprehension systems.

**Broader Impact Concerns:**

The authors may wish to add a broader impact statement regarding the uses of LVC for e.g. security purposes.

**Claims And Evidence:**

No

**Claims Explanation:**

See requested changes.

**Requested Changes:**

1. The causal scoring metric risks missing temporally-distant causal events
The causal influence module evaluates frames only within local windows around sub-scenes chosen for high relevance. This creates a bias that events that occur much earlier in the video may never be evaluated by the causal module. Many long-horizon tasks require reasoning over such precursors (for example, the placement of “Chekhov’s gun” in the beginning of a film may be causally critical to its firing much later).

While I don’t anticipate that this will be solvable with the current framework, it is a meaningful limitation which I recommend that the authors acknowledge and discuss how future versions of MIRA might incorporate long-range causal dependencies. If there are clear ways this can be readily improved, or the if accuracy of causal assessment at long durations can be assessed, I would recommend this be included in the manuscript.


2. Insufficient attribution of improvements
While the paper has used ablation studies to investigate the effectiveness of the different elements of the keyframe composition, the results do not isolate whether performance gains are attributable to the model-feedback loop. An additional ablation study should be done with the model-feedback loop removed. Furthermore, in Table 4 it seems that the vast majority of performance improvement is from relevance sampling, with only very modest gains from adding the causal and summarization pools. This should be acknowledged in the text, as one of the main claims is that *multiple* pools is important for LVC,  and I wonder if the impact of the causal/summarization pools might be improved via changes to the model feedback loop that focus on identifying additional relevant frames for these pools.


3. Limited evaluation of causal claims
The paper uses the term “causal” to refer to the frames extracted from the causal evaluation metric, but does not provide any direct evaluation against ground-truth causal event structure, qualitative examples showing that the causal module retrieves genuinely explanatory frames, or tests involving long-range causal dependencies. As written, the “causal” score reflects semantic consistency under frame ablation, rather than causal reasoning in a strict sense. While this approach is clearly well-motivated by counterfactual reasoning, a more cautious framing, or better yet, additional validation of the causal claims, would strengthen the work.


4. Missing key long-video benchmark: EgoSchema
The evaluation omits EgoSchema, which is a widely used for long-horizon video-language understanding. Even if the authors choose not to evaluate on it, the dataset should be discussed in related work with justification for its exclusion.


Minor concern:
Figure 3 (correct-ratio heatmap) is difficult to interpret. I recommend expanding or clarifying the caption and axis descriptions.

---

> ### Author Response · Authors · 2025-12-29
> **To Reviewer LyCH**
>
> > Q1: The causal scoring metric risks missing temporally-distant causal events.
>
> A1: Thanks for your invaluable advice! We acknowledge that the current neighboring causal modeling approach fails to effectively capture and assess long-range causal dependencies. To address this limitation, we preliminarily explore a convenient improved version, which requires only a minor adjustment to the sub-scene window construction procedure, leaving the remainder of the MIRA framework intact.
>
> Specifically, instead of anchoring each sub-scene window directly at a high-relevance representative frame and expanding temporally forward and backward to form a local window, we consider computing a visual-similarity-based scoring matrix $\mathbf{M}$ over the set of selected representative frames $V_\text{Rep}$. It is worth noting that {$F_t|V_t \in V_\text{Rep}$} have been pre-computed during the scene segmentation stage, enabling the construction of $\mathbf{M}$ via only a single efficient matrix multiplication. Subsequently, given a chosen window anchor $w_c$, we retrieve the top-$2n$ frames with the highest visual similarity to $w_c$ from $\mathbf{M}_{w_c,:}$ as its contextual support, this procedure imposes no constraint on temporal distance between the anchor and its contextual frames, thereby forming a sub-scene window grounded in global causal modeling.
>
> Experimental validation in Table R5 corroborates this enhancement: the average temporal index distance between the window anchor $w_c$ and its contextual counterpart $w_n$ increases significantly—from 8.59 (in the original version) to 146.32–162.82 (in the improved version), demonstrating the enhanced capacity of the improved version to model long-range causal events. We will further explore along the above route in the future.
>
> Table R5: Validation of the causal claims on VideoMME. In contrast, for a keyframe set selected solely based on the relevance metric, the relevance ranking of any frame therein will not exceed 64.
>
> | Sub-scene Window Type | Extraction Mode for $w_n$ | Avg. Relevance Ranking for $w_c$ | Avg. Distance for $[w_n, w_c]$ | Avg. Relevance Ranking for $w_n$ | Avg. Causal Ranking for $w_n$ |
> | - | - | - | - | - | - |
> | neighboring causal modeling | top-1 causal | 268 | 8.59 | 429 | 24 |
> | | all | 268 | 8.59 | 432  | 50 |
> | global causal modeling | top-1 causal | 258 | 162.82 | 425 | 12 |
> | | all | 258 | 146.32 | 410 | 33 |
>
> > Q2: A more cautious framing, or better yet, additional validation of the causal claims, would strengthen the work.
>
> A2: Thanks for this insightful comment! The causal evaluation described in this work exploits the LLM’s inherent capacity for introspection and logical reasoning in the textual domain, deducing causal dependencies among sub-scenes indirectly through a comparison between the LLM-generated sub-scene descriptions and their corresponding ground-truth counterparts. We recognize that the context-masking mechanism adopted herein aligns more closely with the notion of counterfactual intervention and falls short of meeting the formal requirements of causal inference; thus, in the final version, we will plan to rename Causal Influence Score to Counterfactual Influence Score and polish our causal claims.
>
> However, given the complexity of inter-frame content relationships in LVC tasks, along with practical constraints on real-time responsiveness and computational overhead, it is generally infeasible to perform quantitative modeling or generate precise evaluations strictly based on formal causal inference frameworks from traditional machine learning.  We tend to think that the causal assessment methodology proposed in this work serves as a pragmatic and computationally efficient alternative.
>
> In Table R5, we observe that, whether selecting only the context frame with the highest causal score within each sub-scene window or selecting all contextual frames, the average relevance ranking of $w_n$ remains substantially lower than that of $w_c$, indicating that our approach is capable of identifying keyframes beyond mere relevance.  Furthermore, as shown in Table R6, we conduct an additional validation experiment: we employ Qwen3-8B, guided by a specified prompt, to filter VideoMME in order to construct a subset of queries that exhibit strong causal characteristics (several examples of this subset are listed in Table R6).  On this causally enriched query subset, the incorporation of summarization/causality frames yields a 1.9% absolute improvement in answer accuracy.

---

> > ### Author Response · Authors · 2025-12-29
> > **To Reviewer LyCH (Part II)**
> >
> > Table R6: Performance comparison on a causal subset of VideoMME, the prompt we use here is "Analyze the given question to determine if it requires causal reasoning to answer. Output exactly one lowercase word: "causal" if the question requires understanding cause-and-effect relationships, or "non-causal" if it does not. Do not include any extra text.\n Question: <|placeholder|>".
> >
> > | Method  | Accuracy |
> > | - | - |
> > | w/o Summarization/Causality | 64.1 |
> > | w/ Summarization/Causality | 66.0 |
> >
> > **Subset Example:**
> > - **#1 Question:** What would most likely cause Mimi to eat on the floor?
> > - **#2 Question:** Why does the man always speak in a strange way during his date in the latter half of the video?
> >
> > > Q3: An additional ablation study should be done with the model-feedback loop removed. It seems that the vast majority of performance improvement is from relevance sampling.
> >
> > A3: Thanks for this comment. We would like to clarify that the ablation studies in Table 4 are conducted under a specific condition: each input consists of 64 frames in total, with a fixed blending ratio between relevance-based and summarization/causality-based sampling (either 64:0 or 40:24), and without involving the model-feedback loop.
> >
> > The effectiveness of the causal and summarization pools in enhancing MLLMs’ comprehension of long videos is highly dependent on both the blending ratio and the granularity of their application. Figure 3, in fact, illustrates an idealized scenario—namely, where an optimal blending ratio is assigned per video sample. Under such ideal conditions, the full potential of the causal/summarization pools is realized, leading to substantial performance gains, as evidenced by Table R7, Line 4.
> >
> > However, given that current MLLMs and techniques are incapable of determining such precise, sample-adaptive ratios solely from the user query and raw video input, our proposed closed-loop pipeline offers a practical alternative for real LVC circumstance. Specifically, it leverages the MLLM’s intrinsic reasoning capability to enable relatively controllable and dynamic blending ratio assignment, as demonstrated in Table R7, Line 3. Although our approach remains constrained by the inherent limitations of contemporary MLLMs, it consistently outperforms fixed-ratio baselines by more effectively exploiting summarization and causality cues, thereby yielding measurable improvements. This outcome indirectly validates the significance of explicitly incorporating summarization and causality modeling. From this perspective, we posit that once MLLMs attain the level of video-aware reasoning and adaptive allocation capacity envisioned in Figure 3, the full potential of summarization/causality modeling will be unlocked, and the current bottleneck in LVC tasks will be fundamentally addressed.
> >
> > Table R7: Ablation studies for Summarization/Causality, model feedback loop and its ideal performance upper-bound on Qwen2.5-VL-7B.
> >
> > | Relevance | Summarization | Causality | Feedback Loop | Blending Ratio | Granularity | LVB | VideoMME | MLVU |
> > | - | - | - | - | - | - | - | - | - |
> > | w/ | w/o | w/o | w/o | fixed | video pool | 65.3 | 64.8 | 74.4 |
> > | w/ | w/ | w/ | w/o | fixed | video pool | 66.2 (+0.9) | 66.5 (+1.7) | 75.5 (+1.1) |
> > | w/ | w/ | w/ | w/ | relatively dynamic | video pool | 66.9 (+1.6) | 67.0 (+2.2) | 76.3 (+1.9) |
> > | w/ | w/ | w/ | - | dynamic | video sample | 72.8 (+7.5) | 74.6 (+9.8) | 82.2 (+7.8) |
> >
> > > Q4: Missing experimental validation on EgoSchema benchmark.
> >
> > A4: Thanks for pointing it out. EgoSchema, as a pioneering and significant benchmark in the domain of LVC, plays a pivotal role in advancing the field. To further substantiate our claims, we supplement additional experiments on EgoSchema in Table R8. Despite employing a 7B-scale model, MIRA can surpass the previous approach that relies on GPT-4 as the backbone model, demonstrating strong competitiveness of our scheme.
> >
> > In addition, it is worth noting that the duration of each video sample in EgoSchema is 180 seconds. In contrast, benchmarks such as LVB and VideoMME encompass ultra-long videos on the order of one hour. We tend to think that performance gains observed on these more challenging benchmarks may provide stronger evidence of a method’s scalability and robustness for real-world long-video understanding scenarios.
> >
> > Table R8: Performance comparison with previous methods on EgoSchema.
> >
> > | Method    | Model           | Accuracy |
> > |:----------|:----------------|:--------:|
> > | MVU [1]   | Mistral-13B     | 37.6     |
> > | INTP [2]  | Vicuna1.5-7B    | 38.6     |
> > | Vamos [3] | GPT-4           | 48.3     |
> > | MIRA      | Qwen2.5-VL-7B   | 59.0     |

---

> > > ### Author Response · Authors · 2025-12-29
> > > **To Reviewer LyCH (Part III)**
> > >
> > > > Q5: Expanding or clarifying the caption and axis descriptions in Figure 3. Adding a broader impact statement regarding the uses of LVC.
> > >
> > > A5: Thanks for this comment. We will further improve Figure 3 and add the impact statement to the final submitted version. Regarding the impact statement, the MIRA framework has the properties of multi-stage, training-free and closed-loop, which can support LVC tasks in both online and offline scenarios. Calling relevant model processing interfaces locally will further enhance the security of LVC.
> > >
> > > [1] Understanding long videos in one multimodal language model pass, 2024.
> > >
> > > [2] Interpolating video-llms: Toward longer-sequence lmms in a training-free manner, 2024.
> > >
> > > [3] Vamos: Versatile action models for video understanding, ECCV 2024.

---

> > ### Comment · Reviewer_LyCH · 2026-01-12
> > **Further question on causal scoring**
> >
> > Does the global causal approach change the network's performance? This does not seem to be included in the table.

---

> ### Author Response · Authors · 2026-01-14
> **To Reviewer LyCH (Part IV)**
>
> > Q6: Does the global causal approach change the network's performance?
>
> A6: Thanks for this question. As shown in Table R9, we conduct a preliminary evaluation for the global causal modeling approach on VideoMME. Taking into account the temporal continuity of video events, we contend that reference frames exhibiting strong causality generally reside within the contextual neighborhood adjacent to the anchor frame. However, the capacity to model long-range causal dependencies remains equally critical and indispensable for the MIRA framework. Overall, the performance of the two current modeling strategies is roughly comparable. In the future version of MIRA, we will further investigate and refine the global causal modeling scheme.
>
> Table R9: Performance comparison of two causal modeling schemes under different blending ratios (Rel: Sum/Causal) on VideoMME, Qwen2.5-VL-7B.
>
> Sub-scene Window Type / Blending Ratio               | 64:0 | 56:8 | 48:16 | 40:24 | 32:32 | 24:40 | 16:48 | 8:56 | 0:64
> ----------------------------|------|------|-------|-------|-------|-------|-------|------|------
> Neighboring Causal Modeling| 64.8 | 65.9 | 65.7  | 66.5  | 66.2  | 66.4  | 66.7  | 65.9 | 65.6
> Global Causal Modeling     | 64.8 | 65.3 | 65.9  | 65.7  | 65.7  | 66.1  | 66.2  | 66.3 | 65.6

---

### Review · Reviewer_32Gn · 2025-12-17

**Summary Of Contributions:**

MIRA is a training-free keyframe selection framework for long-video question answering with existing multimodal large language models. The paper argues that standard query–frame relevance retrieval tends to over-focus on isolated moments and fails to provide sufficient contextual and explanatory frames for long-horizon video reasoning.

The method constructs three types of frame-selection signals: (1) query–frame relevance based on global and regional CLIP similarity, (2) hierarchical scene and sub-scene summarization where scenes are segmented and representative frames are selected based on aggregated relevance, and (3) a mask-based influence score that estimates how neighboring sub-scenes affect an anchor scene by measuring caption changes under masking. These signals are combined in different ratios to form multiple candidate frame pools.

At inference time, the framework uses simple MLLM feedback to select among these pools, either by prompting the model to classify the query type before sampling or by allowing a “missing information” option that triggers resampling with more frames. Experiments on LongVideoBench, VideoMME, and MLVU with LLaVA-Video and Qwen2.5-VL show consistent over uniform sampling and other training-free baselines.

**Strength:**
- Practical, training-free wrapper: the approach can be applied at test time on top of existing MLLMs without any fine-tuning.

- Clear motivation for multi-view frame selection.

- Reasonable modular design.

**Weakness:**
- The method relies on a long, multi-stage pipeline with **many hand-tuned, interdependent hyperparameters** spanning relevance scoring, scene segmentation, causal windowing, and pool routing.
- How robust is the method with respect to different prompt format? (asking model feedback, etc.)
- The paper’s use of the term “causality” refers to a heuristic sensitivity-to-masking score rather than a formal causal notion, which may be misleading given the lack of an explicit causal model or identifiability assumptions.
- Figure 2 is not reader-friendly. The text color makes it very difficult to read. Furthermore, the middle block does not give a clear difference between sub-scene scores/frame scores and other than that. Overall this figure does not help me understand the paper.

**Audience:**

Yes

**Audience Explanation:**

Test-time inference for VLLM is an important topic in this area

**Claims And Evidence:**

Yes

**Claims Explanation:**

The paper provides details analysis of their method and compare with other SOTA approaches.

**Requested Changes:**

See Weakness. Please address it in your new version.

---

> ### Author Response · Authors · 2025-12-29
> **To Reviewer 32Gn**
>
> > Q1. The method has many hand-tuned, interdependent hyperparameters.
>
> A1: Thanks for this constructive comment. As a training-free framework to LVC, and given that our scheme—like prior works in this domain—relies on specific evaluation criteria to perform keyframe selection from the raw video stream, we acknowledge that this process indeed inevitably involves certain hyperparameters related to scoring and modeling.
>
> We would like to clarify, first, that all relevant hyperparameters are assigned fixed, empirically determined values, without cherry-picking or applying benchmark-specific tricks across different evaluation datasets. Second, among these hyperparameters, we consider the blending ratio between different pool types (i.e., relevance, summarization and causality) as the most critical factor directly influencing the comprehension performance of MLLMs. For this key factor, we present detailed experimental results in Appendix, Table S1 and employ a closed-loop feedback strategy to enable the MLLM to adaptively optimize the blending ratio at test time. Moreover, as insightfully pointed by the reviewer, the influence of the given prompt on model decision-making during the feedback stage warrants further validation, we provide detailed experimental analysis on this aspect in Table R3 with more details explained in our response to Q2.
>
> Table R3: Comparison of different prompt formats in model feedback loop.
>
> | Prompt | LVB | VideoMME | MLVU |
> | :--- | :--- | :--- | :--- |
> | (v1) | **66.9** | 66.7 | 75.8 |
> | (v2) | 66.8 | **66.9** | 75.6 |
> | (v3) | 66.8 | **66.9** | **75.9** |
> | (v4) | **66.9** | **66.9** | **75.9** |
>
> > Q2. How robust is the method with respect to different prompt formats?
>
> A2: Thanks for pointing it out! When prompting the model to provide feedback, we found that the result is generally stable across various prompt formats. We think the reason is that classifying the question into two categories is a relatively simple task for the MLLM as long as the prompt is specific and clear. As shown in Table R3, we list MIRA's performance on three benchmarks after adding the model feedback loop activated by different prompts. It can be observed that our method achieves consistent performance in different prompt formats. The four prompts we tested are as follows:
>
> - Prompt v1: Analyze this question to decide if a single scene from the video / multiple scenes from the video is required to anwser the question. Question: <|placeholder|>. Answer with single word "single/multiple".
>
> - Prompt v2: Analyze the following video question. Can it be answered by looking at only a single scene in the video? Output a single lowercase word: 'single' if yes, 'multiple' if no. Do not include any other text or explanation.\n\nQuestion: <|placeholder|>
>
> - Prompt v3: You are an expert in video question answering. Determine if the following question can be answered by examing a single scene from the video, or if it requires understanding events and relationships across multiple scenes.  Output exactly one lowercase word: 'single' or 'multiple'.  Do not include any other text, punctuation, or explanation. For example:\n Question: What color is the car?  -> single\n Question: What is the woman with the pink hat wearing  -> single\n Question: Why did the person run away? -> multiple\n\n Question: <|placeholder|>
>
> - Prompt v4: You are an expert in video question answering. Determine if the following question can be answered by examining a single scene from the video, or if it requires understanding events and relationships across multiple scenes.  Output exactly one lowercase word: 'single' or 'multiple'. Do not include any other text, punctuation, or explanation.
> For example:\n Question: There is a man with short hair wearing a gray suit. What color are his glasses? -> single\n Question: In the video after feeding the ducks, what did the male protagonist do after riding his bike? -> multiple\n Question: How many times do the woman in red dress appear? -> multiple\n\n' Question: <|placeholder|>.

---

> ### Author Response · Authors · 2025-12-29
> **To Reviewer 32Gn (part II)**
>
> > Q3. The paper’s use of the term “causality” may be misleading given the lack of an explicit causal model or identifiability assumptions.
>
> A3: Thanks for this insightful comment! We recognize that the context-masking-based causal assessment proposed here leans more toward leveraging the idea of counterfactual intervention and employs the LLM’s reasoning capability to conduct an indirect evaluation of causality, rather than strictly satisfying the formal definition of an explicit causal model. We will explicitly clarify this point in the final submitted version.
>
> In Table R4, we further provide a detailed statistical analysis for the sub-scene windows obtained via the current evaluation method. One can find that, across three LVC benchmarks, the relevance-score rankings of the selected causal frames (whether selecting only top-1 causal frame or all of them) are consistently and significantly lower than those of their corresponding window anchor frames, which also confirms that MIRA can effectively exploit contextual information with implicit value that cannot be simply retrieved by relevance scoring.
>
> Table R4:  Relevance and Causality modeling on three benchmarks. In contrast, for a keyframe set selected solely based on the relevance metric, the relevance ranking of any frame therein will not exceed 64.
>
> | | LVB | VideoMME | MLVU |
> | - | - | - | - |
> | Avg. Relevance Ranking for $w_c$ | 219 | 268 | 166 |
> | Avg. Relevance Ranking for $w_n$ (top-1, all) | 310, 311 | 429, 432 | 239, 234 |
> | Avg. Causal Ranking for $w_n$ (top-1, all) | 20, 43 | 24, 50 | 20, 44 |
>
> > Q4. Figure 2 is not reader-friendly.
>
> A4: Thanks for your advice. We will improve the visualization and layout of Figure 2 in the final version and make a clearer distinction among different types of evaluation metrics.

---

### Review · Reviewer_nvXo · 2025-12-17

**Summary Of Contributions:**

The authors introduce a training-free long-video keyframe selection method that improves VQA by mixing global and regional query–frame relevance, hierarchical scene-based representativeness, and a mask-based influence score that measures the impact of removing nearby context on the predicted caption for an anchor sub-scene. The core method builds multiple candidate keyframe pools with different mixture ratios, and uses a lightweight adaptive routing loop with optional handling of "insufficient information" settings to pick a pool in a query-adaptive manner. They report consistent gains on LVB, VideoMME, and MLVU using the LLaVA-Video-7B and Qwen2.5-VL-7B models as backbones.

**Audience:**

Yes

**Audience Explanation:**

The problem of long-video understanding is timely and practically important, with MLLMs being increasingly deployed for video analysis. The core methodology is thoughtfully described and motivated at each step. This clarity is appreciated in making this work accessible and readable to a broader audience while bridging video understanding with a smart retrieve-and-generate framework, which may have insights relevant to several sub-communities (e.g. video-language models, RAG, efficient inference).

**Broader Impact Concerns:**

N/A -- perhaps it would not hurt to include a brief statement on potential misuse risks (e.g. privacy concerns) and how performance can vary across diverse contexts. This could nicely fit with the authors' motivation for the inclusion of the "Missing Information" handling, as well as how this relates to the notions of "causality" and relevance in video understanding and generation through this method.

**Claims And Evidence:**

Yes

**Claims Explanation:**

1. The analysis of diversity in the sampling pool (illustrated by Figure 3) is helpful in showing that different video pools and mixture ratios help across different subsets, and is good motivation to support adaptive routing instead of fixed ratios.
2. There is a thorough comparison against fairly comprehensive baselines, with both training-based and training-free methods and frontier models like GPT-4o.
3. Consistent and clear improvements with two backbone MLLMs models across three benchmarks, although the closed-loop vs open-loop (sampling strategies) comparison in Table 2 is less convincing. The findings do reflect the central contributions claimed, nonetheless.

**Requested Changes:**

1. The “causal” framing does not seem to be empirically supported, since the formulation in Eq. 3 is predictive influence / consistency under masking measure, a proxy for mutual information in caption space rather than actually identifying causality. It does seem a bit strong to term this as a “Causal Influence Score,” to this effect. Ideally, the authors should distinguish what is proven (that this framework / the schemes in Section 3.4 have  utility for QA), from what is intended but not demonstrated (a causal explanation).
2. A table reflecting computational overhead for MIRA would be informative for practitioners, with metrics like wall-clock timing compared with uniform sampling and single-metric retrieval, the number of frames initial sampled in the raw pool for CLIP scoring, masking calls, etc.
3. It would be helpful to compare MIRA with strong training-free baselines on fixed total inference cost rather than just matching frame count. Again, this would inform overhead for practitioners.
4. Please clarify whether Fixed Sampling in Table 2 uses the single best pool from Table S1 or if it is an average across pools.
5. Describe the intuition behind the threshold choices in Appendix A.1; otherwise, these seem a bit arbitrarily chosen. If there is prior literature or some ablations / sensitivity analysis motivating this, please add or cite it.
6. Scaling analysis showing how performance varies as total frame budget increases from 64 to 128 (and maybe to 256) would inform whether the is a change in benefit of this method relative to baselines.
7. “Mask-based causal inference” (page 2 and Figure 2): it would be good to clarify that this refers to masking neighboring frames, not image-level masking.

---

> ### Author Response · Authors · 2025-12-29
> **To Reviewer nvXo**
>
> > Q1: The term “Causal Influence Score” seems a bit strong. It would be good to clarify that “mask-based causal inference” refers to masking neighboring frames, not image-level masking.
>
> A1: Thanks for pointing it out! We concur that the causal evaluation framework introduced in this work is primarily grounded in the notion of counterfactual intervention and utilizes the LLM’s inherent reasoning capacity to infer causal relations in an indirect manner, instead of adhering strictly to the formal requirements of causal inference. In the final version of the manuscript, we will explicitly address this distinction and, in accordance with the reviewer’s suggestion, elaborate more precisely on the notion of “mask-based causal inference.”
>
> > Q2: How performance varies as total frame budget increases from 64 to 128?
>
> A2: Thanks for your insightful comment! We adopt uniform sampling as the baseline and compare it with the multi-view sampling method proposed in this work across multiple keyframe budget settings (32, 64 and 128 frames), as shown in Table R1. It can be observed that, under both sampling schemes, the MLLM’s comprehension performance consistently improves as the number of input frames increases. However, for MLLMs with 7B parameters, once the number of input frames reaches a certain level, the QA accuracy will gradually approach the model’s intrinsic performance upper-bound. Overall, our method consistently outperforms the baseline, and the performance advantage becomes more pronounced when the number of input frames is relatively small.
>
> Table R1: Performance comparison under different keyframe extraction quantities.
>
> | Sampling Strategy | Frames | LVB, Qwen2.5-VL-7B | VideoMME, Qwen2.5-VL-7B |
> |- | - | - | - |
> | Uniform Sampling | 32, 64, 128 | 57.1, 60.0, 61.6 | 60.0, 63.5, 66.4 |
> | Fixed Sampling | 32, 64, 128 | 64.5 (+7.4), 66.2 (+6.2), 66.6 (+5.0) | 65.1 (+5.1), 66.7 (+3.2), 67.6 (+1.2) |
>
> > Q3: Reflecting computational overhead for MIRA, comparing it with uniform sampling and training-free baselines.
>
> A3: Thanks for your constructive advice. In Table R2, we provide a detailed breakdown of the average processing time per video sample across each stage. Specifically, for stages (4) and (5)—the modeling and evaluation stages newly introduced in the MIRA framework, the number of processed frames amounts to only 16% of the total frames sampled from the original video, with a total processing time not exceeding 18 seconds, while delivering an advanced LVC framework capable of integrating multiple perspectives and flexible blending ratios of information types, thereby demonstrating greater potential and scalability in complex long-video understanding scenarios. Therefore, we contend that, compared to previously proposed single-metric retrieval approaches, MIRA achieves a fully acceptable trade-off between LVC capability and computational overhead.
>
> Moreover, it is worth noting that the relevance assessment and sub-scene modeling performed once by MIRA can be reused across various LVC settings with differing frame budgets (e.g. 32, 64, 128 frames), as demonstrated in Table R1, without re-executing the MIRA pipeline. Given this distinctive property, a direct comparison of computational overhead between MIRA and other methods may be inappropriate; since one keyframe retrieval can support MLLM inference under multiple configurations, the average computational cost of MIRA will be further amortized and effectively reduced.
>
> Table R2: The average inference time of video samples on Qwen2.5-VL-7B, LVB. As mentioned in Appendix, here the original frame sampling rate is 1FPS, not less than 128 frames; the maximum values of the sub-scene windows and their lengths modeled by MIRA are 64 and 5, respectively.
>
> | Step | Description | Time / Overhead |
> | - | - | - |
> | (1) | Video Loading | 14.12 s |
> | (2) | Single-metric Retrieval Methods | + feature extraction and relevance calculation time of the specific model |
> | (3) | Scene Segmentation | +0.08 s |
> | (4) | Modeling and Captioning Sub-scene Windows  | +6.12 s |
> | (5) | Causality Evaluation | +11.77 s |
> | — | Avg. Frame Processing Rate for (4)+(5) | 16.00% |
> | (6) | Video Pools Generation | +0.003 s |

---

> > ### Author Response · Authors · 2025-12-29
> > **To Reviewer nvXo (Part II)**
> >
> > > Q4: Describe the the threshold choices in Appendix A.1.
> >
> > A4: Thanks for this comment. Similar to the hierarchical scene segmentation proposed in this work, other prior keyframe extraction methods have also employed techniques involving similarity computation and thresholding, such as clustering [1], spliting [2] and token compression [3]; hyperparameter settings in such training-free sampling approaches is typically unavoidable. Here the empirical setting of the two-level scene segmentation thresholds draws inspiration from the top-down long-video modeling strategy proposed in [1].
> >
> > For first-level scene segmentation, we apply an absolute threshold to coarsely identify potential scene transition locations.  Building upon this, for sub-scene segmentation, we adopt a hybrid strategy that combines a fixed threshold upper-bound with similarity-score quantiles, enabling the batch extraction of video segments with appropriate length and controlled visual redundancy. We empirically verify that our hyperparameter choice works robustly across various benchmarks.
> >
> > > Q5: Whether Fixed Sampling in Table 2 uses the single best pool from Table S1.
> >
> > A5: Thanks for this question. Yes, here Fixed Sampling represents the best performance of a single pool without using the model feedback loop, which we will further clarify in the final version.
> >
> > [1] Videotree: Adaptive tree-based video representation for llm reasoning on long videos, CVPR 2025.
> >
> > [2] Adaptive keyframe sampling for long video understanding, CVPR 2025.
> >
> > [3] APVR: Hour-Level Long Video Understanding with Adaptive Pivot Visual Information Retrieval, 2025.

---

### Decision · Action_Editor_yfRB · 2026-01-26

**Recommendation:** Accept as is

**Audience:**

Yes

**Audience Explanation:**

Long-video understanding is a timely and relevant problem for the TMLR audience, particularly as multimodal LLMs are increasingly used for video analysis.

**Claims And Evidence:**

Yes

**Claims Explanation:**

The claims are generally well supported by the experimental evidence.